Antagonistic antimalarial properties of a methoxyamino chalcone derivative and 3-hydroxypyridinones in combination with dihydroartemisinin against Plasmodium falciparum

Kampoun Tanyaluck 1
Koonyosying Pimpisid 1
Ruangsuriya Jetsada 1
Prommana Parichat 2
http://orcid.org/0000-0003-3913-3057 Shaw Philip J. 2
Kamchonwongpaisan Sumalee 2
Suwito Hery 3
Puspaningsih Ni Nyoman Tri 4
http://orcid.org/0000-0003-4941-2922 Uthaipibull Chairat 2 5
Srichairatanakool Somdet 1 somdet.s@cmu.ac.th
1 Department of Biochemistry, Faculty of Medicine, Chiang Mai University , Chiang Mai , Thailand
2 Medical Molecular Biotechnology Research Group, National Center for Genetic Engineering and Biotechnology (BIOTEC), National Science and Technology Development Agency , Pathum Thani , Thailand
3 Department of Chemistry, Faculty of Science and Technology, Airlangga University , Surabaya , Indonesia
4 Laboratory of Proteomics, University-CoE Research Center for Bio-Molecule Engineering, Universitas Airlangga , Surabaya , Indonesia
5 Thailand Center of Excellence for Life Sciences (TCELS) , Bangkok , Thailand
Braga Erika
Electronic publication date: 2023 Apr 27
Publication date: 2023
Volume: 11
Electronic Location ID: e15187
Received 2022 Dec 15; Accepted 2023 Mar 15
Copyright: © 2023 Kampoun et al.
Copyright year: 2023
Copyright holder: Kampoun et al.
License: This is an open access article distributed under the terms of the Creative Commons Attribution License, which permits unrestricted use, distribution, reproduction and adaptation in any medium and for any purpose provided that it is properly attributed. For attribution, the original author(s), title, publication source (PeerJ) and either DOI or URL of the article must be cited.
License URL: https://creativecommons.org/licenses/by/4.0/

Keywords: Plasmodium, Antimalarial, Artemisinin, Chalcone, Hydroxypyridinone, Drug resistance, Ferredoxin

Funding: Royal Golden Jubilee PhD Program (NSTDA) Thailand Research Fund PHD0234/2558 Faculty of Medicine Endowment Fund, Chiang Mai University 113-2561 This work was funded by the Royal Golden Jubilee PhD Program (NSTDA), the Thailand Research Fund (PHD0234/2558) and the Faculty of Medicine Endowment Fund, Chiang Mai University (Grant number 113-2561). The APC was funded by the Royal Golden Jubilee PhD Program (NSTDA), Thailand Research Fund (PHD0234/2558). The funders had no role in study design, data collection and analysis, decision to publish, or preparation of the manuscript.

==============================
Background

The spread of artemisinin (ART)-resistant Plasmodium falciparum threatens the control of malaria. Mutations in the propeller domains of P. falciparum Kelch13 (k13) are strongly associated with ART resistance. Ferredoxin (Fd), a component of the ferredoxin/NADP+ reductase (Fd/FNR) redox system, is essential for isoprenoid precursor synthesis in the plasmodial apicoplast, which is important for K13-dependent hemoglobin trafficking and ART activation. Therefore, Fd is an antimalarial drug target and fd mutations may modulate ART sensitivity. We hypothesized that loss of Fd/FNR function enhances the effect of k13 mutation on ART resistance.

Methods

In this study, methoxyamino chalcone (C3), an antimalarial compound that has been reported to inhibit the interaction of recombinant Fd and FNR proteins, was used as a chemical inhibitor of the Fd/FNR redox system. We investigated the inhibitory effects of dihydroartemisinin (DHA), C3, and iron chelators including deferiprone (DFP), 1-(N-acetyl-6-aminohexyl)-3-hydroxy-2-methylpyridin-4-one (CM1) and deferiprone-resveratrol hybrid (DFP-RVT) against wild-type (WT), k13 mutant, fd mutant, and k13 fd double mutant P. falciparum parasites. Furthermore, we investigated the pharmacological interaction of C3 with DHA, in which the iron chelators were used as reference ART antagonists.

Results

C3 showed antimalarial potency similar to that of the iron chelators. As expected, combining DHA with C3 or iron chelators exhibited a moderately antagonistic effect. No differences were observed among the mutant parasites with respect to their sensitivity to C3, iron chelators, or the interactions of these compounds with DHA.

Discussion

The data suggest that inhibitors of the Fd/FNR redox system should be avoided as ART partner drugs in ART combination therapy for treating malaria.

Introduction

Artemisinin (ART)-based combination therapy is recommended by the World Health Organization for the first-line treatment of malaria in uncomplicated Plasmodium falciparum infections. ART and its derivatives are biotransformed by the liver to the active metabolite, dihydroartemisinin (DHA) (Fig. 1), which can be activated by iron, resulting in endoperoxide radicals that damage proteins via the formation of covalent adducts. The accumulation of damaged, polyubiquitinated proteins rapidly induces lethal endoplasmic reticulum stress (Bridgford et al., 2018). ART-resistant P. falciparum parasites emerged in Western Cambodia in the 2000s showing a slow clearance phenotype in malaria patients and an increased ring survival rate (Fairhurst & Dondorp, 2016; Tilley et al., 2016). Mutations in the kelch domain of the kelch13 (k13) gene have been established as genetic markers for ART-resistant malaria parasites (Ariey et al., 2014; Straimer et al., 2015); however, ART resistance is modulated by other genetic factors in some resistant parasites that do not contain k13 mutations (Balikagala et al., 2017; Ikeda et al., 2018). A genome-wide association study identified variants in the ferredoxin (fd), apicoplast ribosomal protein S10 (arps10), multidrug resistance protein 2 (mdr2), and chloroquine resistance transporter (crt) genes as additional factors contributing to ART resistance, among which a missense fd mutation (fd-D193Y) was the most frequent variant among resistant parasites (Miotto et al., 2015).

Figure 1 Chemical structures of dihydroartemisinin (DHA), compound 3 (C3), and 3-hydroxypyridinones; deferiprone (DFP), deferiprone-resveratrol (DFP-RVT), and 1-(N-acetyl-6-aminohexyl)-3-hydroxy-2-methylpyridin-4-one (CM1).

Ferredoxin (Fd) is an iron-sulfur (Fe-S) electron carrier protein that functions with ferredoxin NADP+ reductase (FNR) in apicoplast metabolism, particularly isoprenoid biosynthesis, fatty acid desaturation, and heme oxygenation. However, only the isoprenoid biosynthesis pathway is essential during intraerythrocytic developmental stages (Gisselberg et al., 2013; Rohrich et al., 2005). The Fd-D97Y mutant protein (mutated residue corresponding to residue 193 in the full-length pre-processed Fd protein) has an increased binding affinity for FNR. This mutation reduces FNR function; hence, loss-of-function mutations in the Fd/FNR system may contribute to ART resistance (Kimata-Ariga & Morihisa, 2021). Isoprenoids are protein prenylation substrates, and prenylated proteins are required for vesicular transport to the digestive vacuole, including hemoglobin transport, in which K13 is thought to play an important role (Birnbaum et al., 2020; Mok et al., 2021). Hemoglobin degradation products are required for ART activation, and defects in hemoglobin trafficking cause ART resistance (Mukherjee et al., 2022). Therefore, loss of Fd/FNR function caused by fd mutation or chemical inhibition may enhance the effect of the k13 mutation on the hemoglobin trafficking pathway, resulting in enhanced ART resistance. However, fd/fnr mutations are constrained, because this redox system is essential (Swift et al., 2022). Moreover, Fd/FNR is a validated antimalarial drug target (Qidwai et al., 2014; Suwito et al., 2014). The interaction between recombinant PfFd and PfFNR proteins is inhibited in vitro by methoxyamino chalcone derivatives, of which the most potent compound, C3 (Fig. 1), exhibits 50% inhibition at 100 µM (Suwito et al., 2014).

In addition to Fd, iron is a cofactor for enzymes such as aconitase, oxidoreductases, and ribonucleotide reductase. Hemoglobin, myoglobin, and cytochromes contain an iron-prosthetic heme group. Iron also acts as a chemical catalyst for the generation of reactive oxygen species (ROS) via the Fenton and Haber–Weiss reactions. Iron chelators, such as desferrioxamine (DFO), 1,2-dimethyl-3-hydroxypyridin-4-one, deferiprone (DFP), and deferasirox (DFX) are clinically used for the treatment of patients with iron overload. Importantly, DFO, DFP, DFX, and other iron chelators, including 1-(N-acetyl-6-aminohexyl)-3-hydroxy-2-methylpyridin-4-one (CM1) and deferiprone-resveratrol hybrid (DFP-RVT) possess antimalarial activity by interfering with iron uptake, depleting the intraerythrocytic labile iron pool in parasites and expediting the phagocytosis of ingested parasites (Cabantchik, Moody-Haupt & Gordeuk, 1999; Chuljerm et al., 2021; Gordeuk, Thuma & Brittenham, 1994; Maneekesorn et al., 2021; Thipubon et al., 2015). Some antimalarial drugs such as quinine, mefloquine, and artesunate are antagonized by DFP (Pattanapanyasat et al., 2001).

In combination therapy, two different drugs are used to increase the efficacy of treatment, and the interaction between these drugs can be assessed by isobologram analysis (Huang et al., 2019). Drugs with a connection in their mechanisms of action, such as those acting on the same metabolic pathway, can exhibit synergistic or antagonistic interactions. Iron chelators are antagonistic to ART and related compounds (Antoine et al., 2014; Suwito et al., 2014), which can be explained by the iron-dependent activation of ART (Meshnick et al., 1993; Pattanapanyasat et al., 2001; Xie et al., 2016; Klonis, Creek & Tilley, 2013). Although the putative loss of function in the fd-D193Y mutant is associated with ART resistance, it does not affect the survival of P. falciparum parasites exposed to DHA (Kampoun et al., 2022; Stokes et al., 2021). Therefore, it is unclear whether the parasite genetic background, in particular, the allelic status of the fd and k13 genes, affects the antimalarial potency of compounds targeting Fd/FNR and/or the interaction of these compounds with DHA. We hypothesized that the antimalarial potency of the chalcone derivative C3 and its interaction with DHA are affected by the genetic background of the parasite. Therefore, we evaluated the potencies of C3, DFP, DFP-RVT, and CM1, and the combinations of these compounds with DHA against P. falciparum wild-type (WT) and genome-edited parasites with k13 and fd mutations. The iron chelators were used as reference ART antagonists for assessing the interaction of C3 with DHA. This study provides information regarding DHA interactions with compounds that target Fd/FNR.

Materials and Methods

Chemicals and reagents

D-Sorbitol, dimethylsulfoxide (DMSO), hydroxyethylpiperazine ethane sulfonic acid (HEPES), RPMI 1640 medium, and SYBR® Green I were purchased from Sigma-Aldrich Chemical Company (St. Louis, MO, USA). Milli®-deionized water (DI) was purchased from Merck KGaA (Darmstadt, Germany).

Drug and compounds

DHA (item No. 19846, MW = 284.4 g/mol) was purchased from Cayman Chemical Company (Ann Arbor, MI, USA). C3 compound (MW = 254.1183 g/mol) was synthesized by Dr. Hery Suwito as previously described (Suwito et al., 2014). DFP (MW = 139 g/mol) was purchased from Sigma-Aldrich (St Louis, MO, USA). DFP-RVT (MW = 340 g/mol) was kindly supplied by Dr. Yongmin Ma, School of Pharmaceutical and Chemical Engineering, Taizhou University, Taizhou, People’s Republic of China, which was synthesized by Xu and colleagues (Kampoun et al., 2022). 1-(N-Acetyl-6-aminohexyl)-3-hydroxy-2-methylpyridin-4-one or CM1 (MW = 266 g/mol), which is a DFP analog and an orally active bidentate iron chelator, was synthesized and kindly supplied by Dr. Kanjana Pangjit, College of Medicine and Public Health, Ubon Ratchathani University, Ubon Ratchathani, Thailand (Pangjit et al., 2015; Thipubon et al., 2015).

Stock solutions of compounds

For the parasite growth inhibition assay, stock solutions of DHA, C3 compound, and DFP-RVT were prepared with 1% (w/v) DMSO as the solvent, whereas DFP and CM1 were dissolved in sterile DI water at 1,000 times the highest dose tested. Ten two-fold serial dilutions were prepared from the stock solutions. For the drug combination study, the drug or compounds were first dissolved in solvents at 2,000 times the highest dose tested. Five two-fold serial dilutions of the stock solutions were prepared. Stock solutions were sterilized using a sterile syringe filter (hydrophilic polyvinylidene difluoride membrane, 0.22 µm pore size, Sigma-Aldrich Chemicals Company, St. Louis, MO, USA) and stored at −20 °C.

Plasmodium falciparum culture and synchronization

The 3D7 parasite line and the transgenic 3D7 parasite lines in this study, which were established in our earlier study, were cultured and maintained in the same conditions in which the parasites grew at the same rate in these culture conditions (Kampoun et al., 2022). Briefly, the parasites were cultured in complete RPMI 1640 medium pH 7.4 containing 2 mM L-glutamine, 25 mM HEPES, 2 g/L NaHCO3, 27.2 mg/L hypoxanthine, and 0.5% Albumax II using human O+ blood group erythrocytes (2–4% hematocrit). Parasite cultures were incubated in 90% N2, 5% CO2, and 5% O2 at 37 °C and synchronized to the ring stage using 5% D-sorbitol treatment. Parasite developmental stage and viability were routinely assessed by microscopic examination of Giemsa-stained thin smear films.

Plasmodium falciparum growth inhibition assays

Before use, stock solutions of the compounds were 100× diluted in a complete RPMI 1640 medium. Ten microliters of each dilution were added to each well of a black 96-well plate. Synchronized ring-stage P. falciparum was cultured in complete RPMI 1640 medium at 1% parasitemia and 2% hematocrit, which was dispensed at 90 µL/well to perform the treatments. The parasite was treated in technical duplicates with various doses of the test compound in a black 96-well plate, while a mock treatment (parasite suspension in culture medium with 0.1% DMSO) was also performed. The mock treatment was assigned as 100% parasite growth. The treated parasites were incubated under standard culture conditions for 48 h and the parasite survival rate was determined using the malaria SYBR® Green I-based fluorescence (MSF) assay as described previously (Maneekesorn et al., 2021).

For the MSF assay, 0.2 µL of SYBR® Green I was diluted in 1 mL of lysis buffer solution containing 20 mM Tris-HCl, 5 mM EDTA, 0.008% (w/v) saponin, and 0.08% (v/v) Triton X100. SYBR® Green I solution (100 µL) was added to each well of a 96-well black microplate (Corning® Product number CLS 3601, polystyrene flat-bottom type, Merck KGaA, Darmstadt, Germany). The plate was mixed using a MixMate® Eppendorf machine (Eppendorf SE, Hamburg, Germany) at 1,000 rpm for 30 s and incubated for 1 h in the dark at room temperature. Fluorescence intensity (FI) was measured using a fluorescence multi-well plate reader (Beckman Coulter AD340C; Beckman Coulter Inc., Brea, CA, USA) with excitation and emission wavelengths of 485 and 530 nm, respectively. The concentration of the test compound with 50% inhibition of growth (IC50) was calculated using the drc R package (Ritz & Streibig, 2005).

Combination treatment against P. falciparum growth

Working solutions of the drug or compound were mixed in a 1:1 ratio (5 µL each) in a checkerboard manner in a black 96-well plate, in which the concentration of one drug or compound was fixed while the concentrations of the other were varied. The parasite suspension (1% rings, 2% hematocrit) was then added to each well (100 µL/well in total) and cultured for 48 h under standard conditions. The parasite growth was determined using the MSF assay as described above. Data were obtained from at least three independent replicates, with two technical replicates per experiment.

Isobologram analysis of drug combination

The interaction of DHA with Fd-FNR inhibitor or iron chelator was assessed by isobologram analysis of IC50 values. The FIC indices were calculated from the ratio of the IC50 value obtained from the combination treatment to that obtained from the single compound treatment. The ΣFIC for the combination is the summation of an individual drug or compound. The FIC value was calculated as follows:

ΣFIC=[(IC50 of A in the mixture)÷[IC50 of A alone)]+[(IC50 of B in the mixture)÷[IC50 of B alone)].

The FICs of the combinations from at least three individual experiments were used to calculate the mean values of ΣFIC. Interactions were assigned as synergistic (ΣFIC < 0.9), additive (0.9 < ΣFIC < 1.1), or antagonistic (ΣFIC > 1.1), with 1.1 < ΣFIC < 1.2 representing slight antagonism, 1.2 < ΣFIC < 1.45 representing moderate antagonism, 1.45 < ΣFIC < 3.3 representing antagonism, 3.3 < ΣFIC < 10 representing strong antagonism, and ΣFIC > 10 representing very strong antagonism (Chou, 2006; Crisafulli et al., 2021).

Statistical analysis

Data for each experiment were obtained from at least three independent replicates. The IC50 values of the mutant parasites were compared with those of the parental 3D7 parasite using drc R package by pair-wise t-tests with Bonferroni–Holm post-hoc correction of the P-values, in which P < 0.05 was considered significant (Ritz & Streibig, 2005). The mean ΣFICs of fd-D193Y parasites were statistically compared with the mean ∑FIC of fd wild-type parasites using GraphPad Prism 8.3.0 software by unpaired t-test with Welch’s correction.

Results

Single compound sensitivity test of P. falciparum

The growth inhibitory properties of DHA, C3, DFP, DFP-RVT, and CM1 were assessed in P. falciparum parental strain 3D7 and the transgenic parasites fdD193Y_3D7, k13C580Y_3D7, and k13C580YfdD193Y_3D7 (Kampoun et al., 2022). The potency of the compounds varied as follows: DHA > DFP-RVT > C3 > DFP ≈ CM1, in which the IC50 95% confidence intervals for DHA, DFP-RVT, C3, DFP, and CM1 were 4–5 nM, 8–10 µM, 21–25 µM, 32–41 µM, and 32–38 µM, respectively. However, the IC50 values of the compounds were not significantly different between the 3D7 parental strain and the transgenic strains with k13 and/or fd mutations (Fig. 2).

Figure 2 Dose-response analysis of growth inhibition for DHA, C3, DFP, DFP-RVT, and CM1.

P. falciparum parental parasite 3D7 and transgenic parasites fdD193Y_3D7, k13C580Y_3D7, and k13C580YfdD193Y_3D7 were treated with varying doses of each compound. (A) The dose-response graph plots for IC50 are shown. Data were obtained from at least three independent experiments and dose-response models were fitted to the data. The X axis indicates parasitemia as a term of percentage of non-treatment (100%), while the Y axis indicates compound concentration in molar units (M). The IC50 values acquired from the graph are annotated at the upper right corner. (B) The bar graphs represent a statistical comparison of IC50 values between the different parasite lines. The X axis indicates the IC50 value in micromolar units (µM). The IC50 values and the associated confidence intervals are shown. P-values from pair-wise t-tests with Bonferroni-Holm post-hoc correction comparing the IC50 of each transgenic parasite to that of the parental 3D7 parasite are shown above the IC50 values. A 0.05 P-value is used to indicate any difference. FMT = fdD193Y mutation, KMT = k13C580Y mutation.

Test of pharmacological interaction between DHA and test compounds in P. falciparum

The interactions between DHA and the test compounds were assessed as the fractional inhibition concentration (ΣFIC) index, and the mean ΣFICs of the combinations were compared between fd wild-type (fdWT) and fd-mutated (fdMT) parasites using Welch’s t-test (Fig. 3). An antagonistic effect was indicated when ΣFIC > 1.1, an additive effect was indicated when 0.9 < ΣFIC < 1.1, and synergism was indicated when ΣFIC < 0.9, using the criteria established by previous studies of antimalarial interactions (Chou, 2006; Crisafulli et al., 2021). For the DHA and C3 combinations, the mean ΣFICs were moderately antagonistic, which were 1.24, 1.32, 1.21, and 1.23 in the parental 3D7, fdMT_3D7, k13MT_3D7, and k13MTfdMT_3D7 mutants, respectively. Moreover, no significant differences in mean ΣFIC were observed between fdWT and fdMT parasites. All combinations of DHA with iron chelator compounds were antagonistic, although no significant difference was observed between the fdWT and fdMT parasites.

Figure 3 Isobologram testing the interaction between DHA and test compounds (candidate Fd-FNR inhibitor C3 or iron chelators DFP, DFP-RVT, and CM1) against parental 3D7 and transgenic FMT_3D7 (fd mutant, k13 wild-type), KMT_3D7 (fd wild-type, k13 mutant), and KMTFMT_3D7 (fd mutant, k13 mutant) P. falciparum.

The isobologram was generated by plotting the mean FIC values of DHA against the mean FIC values of C3 or the iron chelators. Data were obtained from at least three independent experiments; mean values are shown and error bars represent SD. The ΣFIC values for each combination are shown next to the isobologram. Statistical comparisons of ΣFIC values between fd-mutated and fd wild-type parasites were performed with a pairwise t-test, for which the P value is given at the lower right corner of each graph. FMT = fdD193Y mutation, KMT = k13C580Y mutation. ΣFIC < 0.9 indicates synergistic, 0.9 < ΣFIC < 1.1 indicates additive, and ΣFIC > 1.1 indicates antagonism.

Discussion

ART resistance is modulated by several factors, including the genetic background and pharmacological interactions with other antimalarials. For example, the crt and mdr1 genes affect parasite response to Ca2+/Na+ channel blockers and their interactions with DHA (Eastman et al., 2016). Here, we investigated the interaction of C3, an inhibitor of the PfFd-PfFNR protein interaction, with DHA in parasites with different fd and k13 genetic backgrounds. C3 demonstrated moderate antagonistic activity toward DHA. Our hypothesis that inhibition of the Fd/FNR redox system in the apicoplast by C3 antagonizes DHA is in line with a recent report that other apicoplast-targeting antimalarials also antagonize DHA (Crisafulli et al., 2021). It should be noted that the degree of DHA antagonism by C3 cannot be directly compared with apicoplast targeting drugs reported previously (Crisafulli et al., 2021) owing to substantial methodological differences. Defects in the key apicoplast functions of isoprenoid metabolism and protein prenylation are thought to antagonize ART by the consequent reduction in hemoglobin trafficking (Crisafulli et al., 2021). Reduction in hemoglobin trafficking reduces the level of free heme from hemoglobin digestion, which is the main catalyst for ART activation. These findings caution against the use of ART partner drugs that inhibit Fd-FNR interaction and other apicoplast targets, which may reduce ART efficacy in malaria chemotherapy.

The antimalarial potency of the Fd-FNR inhibitor C3 was modest (IC50 95% confidence interval 23–25 µM) and did not differ between 3D7 and genome-edited parasites, including those with the fd D193Y mutation. From the study of C3 docked with Fd, C3 was predicted to bind Fd at a site distant from the D193Y mutation (Suwito et al., 2014), which might explain why no difference in antimalarial potency was observed between the fdWT and fdMT parasites. In contrast, the antimalarial effect of C3 could be due to the inhibition of targets other than Fd, as the antimalarial IC50 is markedly lower than the concentration of C3 required for 50% inhibition of the interaction between recombinant Fd and FNR proteins (100 µM) (Suwito et al., 2014). The potential for promiscuous targeting by the chalcone derivative C3 is plausible because other chalcone derivatives are known to be promiscuous-targeting antimalarials, such as licochalcone A, the first identified anti-malarial chalcone derivative, which inhibits P. falciparum mitochondrial complexes II and III, and possibly the erythrocyte membrane (Mi-Ichi et al., 2005; Ziegler et al., 2004). Moreover, phenylurenyl chalcones, which are effective against quinoline-resistant P. falciparum, inhibit parasite cysteine proteases involved in hemoglobin degradation and hemozoin formation (Domínguez et al., 2005), and chalcone-chloroquine-based hybrids exert antimalarial activity by inhibiting hemozoin formation (Guantai et al., 2011; Sashidhara et al., 2012). In this study, the tested iron chelators showed modest antimalarial activity, similar to previous reports (Chuljerm et al., 2021; Ferrer et al., 2012; Maneekesorn et al., 2021; Thipubon et al., 2015). No differences in sensitivity to DHA were observed among the parasites tested, although it should be noted that the ART-resistance phenotype manifested in k13 mutants is not detectable by the growth inhibition assay employed in this study (Ariey et al., 2014).

Importantly, the interactions of C3 with DHA in different parasite backgrounds were moderately antagonistic; hence, inhibition of the Fd-FNR interaction by C3 could antagonize DHA. Nevertheless, there was no significant difference between the fdWT and fdMT parasites, even in the parasite that contains both fd and k13 mutations. Therefore, the antimalarial mode of action of C3 remains unclear. As expected, the iron chelators antagonized the antimalarial activity of DHA. As no significant differences in the mean ΣFICs between fdWT and fdMT parasites were observed, it can be concluded that this fd mutation does not affect the antagonistic interaction between DHA and iron chelators. Presumably, fd mutations associated with the loss of Fd function have a negligible impact on the iron pool targeted by iron chelators. The effect of the fd mutation on the interaction with DHA might be smaller than that of C3, such that it could not be detected by our experimental approach. Alternative approaches are required detect subtle interactions between DHA and apicoplast-targeting drugs. Apicoplast-targeting drugs typically exhibit a delayed-death antimalarial effect (Kennedy et al., 2019), such that interactions with the much faster-acting but short-lived DHA may not be detectable in assays with treatment conducted over one parasite replication cycle, as used in our study. To account for the different rates of killing, interaction assays can be performed by treating with apicoplast inhibitors for up to 72 h before pulsing with DHA (Crisafulli et al., 2021). Another possibility that should be considered is that the fd-D193Y mutation has no detectable effect on DHA interaction with other drugs, but is present at a high frequency among ART-resistant parasites because of positive selection to compensate for loss of fitness caused by k13 mutation, or genetic hitchhiking with k13 resistance mutations.

Conclusions

The putative Fd-FNR inhibitor C3 demonstrated antimalarial activity comparable to that of iron chelators DFP, DFP-RVT, and CM1. No difference in sensitivity was observed among parasites with mutations in the fd and k13 genes compared with the parental P. falciparum 3D7 strain for any compound tested. In combination with DHA, C3 showed moderate antagonistic interactions, and iron chelators were antagonistic. Other approaches are required to understand the effect of the fd mutation on Fd function and how this affects the interaction of C3 with ART.

Supplemental Information

Supplemental Information 1 Raw Data for Figs. 1–3.

(A) Fd int. IC50 values. (B) Fe chelators (IC50 values). (C) DHA (IC50 values). (D) DFP (IC50 values). (E) DFP-RVT (IC50 values). (F) CM1 (IC50 values).

Click here for additional data file.

We thank Dr. Yongmin Ma and Dr. Kanjana Pangjit for supplying DFP-RVT and CM1 compounds, respectively. We also thank the blood donors who supplied blood for P. falciparum culture.

Additional Information and Declarations

Competing Interests

Author Contributions

Data Availability

The authors declare that they have no competing interests.

Tanyaluck Kampoun conceived and designed the experiments, performed the experiments, analyzed the data, prepared figures and/or tables, authored or reviewed drafts of the article, and approved the final draft.

Pimpisid Koonyosying conceived and designed the experiments, performed the experiments, analyzed the data, prepared figures and/or tables, and approved the final draft.

Jetsada Ruangsuriya analyzed the data, prepared figures and/or tables, and approved the final draft.

Parichat Prommana performed the experiments, analyzed the data, prepared figures and/or tables, and approved the final draft.

Philip J. Shaw analyzed the data, prepared figures and/or tables, authored or reviewed drafts of the article, and approved the final draft.

Sumalee Kamchonwongpaisan performed the experiments, analyzed the data, prepared figures and/or tables, authored or reviewed drafts of the article, and approved the final draft.

Hery Suwito conceived and designed the experiments, analyzed the data, prepared figures and/or tables, henry Suwito contributed to a validation of DHA toxicity, and approved the final draft.

Ni Nyoman Tri Puspaningsih conceived and designed the experiments, analyzed the data, prepared figures and/or tables, ni Niyoman Tri Puspaningsih contributed to a validation of compound 3, and approved the final draft.

Chairat Uthaipibull conceived and designed the experiments, prepared figures and/or tables, authored or reviewed drafts of the article, and approved the final draft.

Somdet Srichairatanakool conceived and designed the experiments, analyzed the data, prepared figures and/or tables, authored or reviewed drafts of the article, and approved the final draft.

The following information was supplied regarding data availability:

The raw data is available in the Supplemental File.

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
