# Peer review of "Antagonistic antimalarial properties of a methoxyamino chalcone derivative and 3-hydroxypyridinones in combination with dihydroartemisinin against Plasmodium falciparum"

_PeerJ, doi:10.7717/peerj.15187_

## Round 0.1 · original submission · Major Revisions

The review process is now complete, and four thorough reviews from highly qualified referees are included at the bottom of this letter. Some concerns must be considered in your resubmission. I strongly agree with reviewer #3 and emphasize that the author must dedicate to answer the points raised with the utmost precision and to make all those reconsiderations in the manuscript to be submitted.

Reviewer 1 ·

Basic reporting

No comment

Experimental design

No comment

Validity of the findings

No comment

Additional comments

The manuscript by Tanyaluck Kampoun et al., reports on an in vitro investigation of the behaviour of chalcone, hydroxypyridinon and deferiprone-resveratrol towards wild-type P.falciparum parasites. Malaria, the most prevalent parasitic disease in the world, is caused by the apicomplex protozoan of the Plasmodium genus. The reason for such high morbidity and mortality stems from the fact that P. falciparum, the most dangerous of the malarial parasites that infect humans, and the most effective malaria vector and the most difficult to control (Anopheles gambiae) is widespread in Africa continent. The best available treatment, particularly for P. falciparum malaria, is artemisinin-based combination therapy. However P. falciparum resistance to ATCs has emerged, threatening global malaria control and elimination. So, new malaria drugs with antiplasmodial mechanisms different from those of available medicines are urgently needed. The authors present an interesting work about new substances or drugs demonstrating activity or not against strains of plasmodium falciparum.
In conclusion, the encouraging results obtained in this study open the way for further in vitro and in vivo investigations.
The manuscript can be accepted in the present form.

Annotated reviews are not available for download in order to protect the identity of reviewers who chose to remain anonymous.

·

Basic reporting

This manuscript by Srichairatanakool and colleagues provides a careful study of the interaction of methoxy amino chalcone C3, and three different iron chelators, deferiprone (DFP), 1-(N-acetyl-6-aminohexyl)-3-hydroxy-2-methylpyridin-4-one (CM1) and a deferiprone-resveratrol hybrid (DFP-RVT), against the artemisinin (ART) derivative, DHA, in assays of killing of parental 3D7 and transgenic FMT_3D7 (fd mutant, k13 wildtype), KMT_3D7 (fd wildtype, k13 mutant), and KMTFMT_3D7 (fd mutant, k13 mutant) P. falciparum.

Experimental design

The authors examined the antiplasmodial activity of DHA, chalcone (C3) and iron chelators against wildtype (WT) P. falciparum parasites and those with k13 and fd mutations. There was no difference in action of DHA against WT and K13 mutant parasites. This at first seems surprising given that K13 mutation is known to be associated with resistance to DHA. However, previous studies have employed 3- to 6-h exposure assays in the ring stage of infection, whereas the assay format used here was a 48-h exposure assay. No difference is expected under these conditions. The authors note this design fault in the Discussion; but they do not explain why conditions were chosen that they are not ideal for monitoring subtle difference in susceptibility to inhibitors.

Similarly, the authors looked for interactions between the different inhibitors over a 48-h exposure period. Again, this is not ideal as subtle interactions may be missed. They observed an antagonistic interaction between the iron chelators and DHA, in agreement with previous reports. However, they observed no difference between the wildtype and mutant parasites. For the DHA and C3 combinations, a moderate antagonism was observed, but again no differences were observed between the wildtype and mutant parasites.

Validity of the findings

The authors state that methoxy amino chalcone (C3) is an inhibitor of the ferredoxin (Fd)/ Fd-NADP+ rReductase (Fd/FNR) interaction (50% inhibition at 100 µM), based on a previous in vitro study that involved one of the authors (Ni Nyoman Tri Puspaningsih). This is an interesting suggestion. However, other studies reported that chalcones kill malaria parasite in culture with an IC50 value in the low uM range (See doi.org/10.3390/molecules
27207062 for review). Thus, the weak in vitro activity of chalcones appears inconsistent with inhibition of the Fd/FNR interaction being the main mechanism of action. Indeed, some of those authors provided evidence that chalcones exert their activity by the inhibition of hemozoin formation (eg DOI: 10.1016/j.bmc.2012.03.011).

In the Discussion, the authors state that their observation that C3 antagonizes ART is in line with “..a recent report that other apicoplast-targeting antimalarials also antagonize ART [31].” The experiments in the Crisafulli et al manuscript (ref 31) used a very different format, in which the apicoplast inhibitor was applied in the first cycle and antagonism with DHA was assayed in the second cycle (when the delayed death inhibitors have had an effect on apicoplast metabolism).

In the Discussion the authors discuss the possibility that C3 may hit a target other than Fd/ FNR. It would be useful to include consideration of the possibility that C3 inhibits hemozoin formation – a target that would be consistent with the observed antagonism with DHA.

The authors should discuss the possibility that the fd mutation provides a fitness advantage, helping to compensate for the fitness cost associated with the K13 mutation.

Reviewer 3 ·

Basic reporting

The main issue with this manuscript that needs to be addressed is the volume of work. Whilst conducted appropriately, it is insufficient as it stands. This body of work can be expanded further to elucidate the mechanism of action of the C3 compound and how the Fd mutation modulates artemisinin resistance, as well as several other suggestions the authors had. Appropriate language to the field has been used throughout the manuscript. A few corrections are necessary regarding the English language, in order for the audience to not misinterpret the writing, e.g. line 58: ‘therefore’ instead of ‘nonetheless’. The introduction and background to the work have been sufficiently covered. Additional notes:

• Introduce C3 as the compound of interest in the abstract.
• The abstract conclusion needs to be clarified; stating what was expected may help with understanding line 64-65 (was it surprising or expected?). DHA combined with C3 or iron chelators were antagonistic towards each other, not Pf growth? If so, reword that sentence.
• In the introduction, what is the role of Kelch13, if known?
• Lines 97-99, this is in vitro in Pf? 100uM is too high for an effective compound, is it not?

Experimental design

The knowledge gap and hypothesis have been explained. A statement should be added at the end of the introduction about what the outcome will contribute. Methods have been described with sufficient detail.

Validity of the findings

The data is appropriate (replicates, controls, statistics). Figure 2 needs better resolution (make size of graphs and font the same as Figure 3). The figure 2 and 3 legends need further descriptions so that the figures alone can be understood, e.g. what the axes represent, the statistical measure shown, sample numbers, statistical tests used, etc. Figure 3 needs definitions for what the FIC ranges mean (explained in the results section). See below for specific statements/questions. The conclusions have been clearly stated.

Results section starting at line 218:
• Indicate the IC50s (ranges) for the compounds in the written section.
• Indicate that the IC50 for DHA in the k13 mutant is the same.
• Indicate that the IC50 for C3 in the Fd mutant is the same.

Results section starting at line 224:
• Line 230-231, include actual FIC value.

Discussion:
• Comment on the high IC50s for C3 and the iron chelators. Are compounds with high IC50s of any use?
• Which assay is required to see the expected decreased DHA sensitivity in the k13 mutant? Can this assay be conducted?
• Comment on the use of 48hr assays. How might longer assays affect the results? Has a longer time point to capture two cycles been conducted?
• Line 266: suggest alternative approaches.

Reviewer 4 ·

Basic reporting

Introduction Line 94-96: This mutation suppresses FNR function; hence, loss-of-function mutations in the Fd/FNR system could contribute to ART resistance.- Lacks clarity


Your 'Discussion' needs more information on similar studies.

Experimental design

Methods described with sufficient detail & information to replicate.

Validity of the findings

No comment

---

## Round 0.2 · accepted · Accept

The authors have satisfactorily addressed all review comments and made the necessary changes to the manuscript.

·

Basic reporting

.

Experimental design

.

Validity of the findings

.

Additional comments

The authors have satisfactorily answered my queries.